# Older adults' attitudes towards deprescribing and medication changes: a longitudinal sub-study of a cluster randomised controlled trial

Katharina Tabea Jungo [1,2] Kristie Rebecca Weir [1,3] Damien Cateau [4,5] Sven Streit[1]

KTJ and KRW are joint first authors.

For numbered affiliations see end of article.

**Correspondence to**
Dr. Katharina Tabea Jungo;
katharina.jungo@protonmail.com

## ABSTRACT

**Objective** To investigate the association between older patients' willingness to have one or more medications deprescribed and: (1) change in medications, (2) change in the appropriateness of medications and (3) implementation of prescribing recommendations generated by the electronic decision support system tested in the 'Optimising PharmacoTherapy In the Multimorbid Elderly in Primary CAre' (OPTICA) trial.

**Design** A longitudinal sub-study of the OPTICA trial, a cluster randomised controlled trial.

**Setting** Swiss primary care settings.

**Participants** Participants were aged ≥65 years, with ≥3 chronic conditions and ≥5 regular medications recruited from 43 general practitioner (GP) practices.

**Exposures** Patients' willingness to have medications deprescribed was assessed using three questions from the 'revised Patient Attitudes Towards Deprescribing' (rPATD) questionnaire and its concerns about stopping score.

**Measures/analyses** Medication-related outcomes were collected at 1 year follow-up. Aim 1 outcome: change in the number of long-term medications between baseline and 12 month follow-up. Aim 2 outcome: change in medication appropriateness (Medication Appropriateness Index). Aim 3 outcome: binary variable on whether any prescribing recommendation generated during the OPTICA medication review was implemented. We used multilevel linear regression analyses (aim 1 and aim 2) and multilevel logistic regression analyses (aim 3). Models were adjusted for sociodemographic variables and the clustering effect at GP level.

**Results** 298 patients completed the rPATD, 45% were women and 78 years was the median age. A statistically significant association was found between the concerns about stopping score and the change in the number of medications over time (per 1-unit increase in the score the average number of medications use was 0.65 higher; 95% CI: 0.08 to 1.22). Other than that we did not find evidence for statistically significant associations between patients' agreement with deprescribing and medication-related outcomes.

**Conclusions** We did not find evidence for an association between most measures of patient agreement with deprescribing and medication-related outcomes over 1 year.

**Trial registration number** NCT03724539.

## STRENGTHS AND LIMITATIONS OF THIS STUDY

⇒ A longitudinal sub-study (n=298 patients) of the Optimising PharmacoTherapy In the Multimorbid Elderly in Primary CAre (OPTICA) trial—a cluster randomised controlled trial conducted in Swiss primary care settings.

⇒ Few studies have explored the association between medication-related outcomes after 1 year and patients' hypothetical willingness to have their medications deprescribed as measured by a self-report questionnaire.

⇒ The medication-related outcomes included not only the number of medications between baseline and 12 month follow-up, but also medication appropriateness (as measured by the Medication Appropriateness Index).

⇒ The longitudinal study design allowed for a clear temporal distinction between patients' willingness to have medications deprescribed assessed at baseline and the medication-related outcomes after 1 year.

⇒ Older adults agreeing to participate in the OPTICA trial could have had a higher willingness to have one or more of their medications deprescribed.

## INTRODUCTION

Globally, there is increasing focus on polypharmacy in the older population. Up to 50% of older adults aged 65 years and above take one or more inappropriate medication,[1] which has been associated with harmful effects on health outcomes and quality of life.[2 3] In older patients with multiple chronic conditions (multimorbidity), the percentage is even higher.[4] A medication is considered *inappropriate* when potential harms outweigh potential benefits in the individual.[5] One strategy to mitigate against inappropriate medication use is deprescribing, the process of reducing or stopping medications that lack benefit or may cause harm.[6] However, implementing deprescribing decisions in clinical practice is challenging.

---

**Box 1  Attitudes towards deprescribing: rPATD questions[16]**

**Global question:**
'If my doctor said it was possible, I would be willing to stop one or more of my regular medicines'.

**Appropriateness questions:**
'I would like to try stopping one of my medicines to see how I feel without it'.
'I would like my doctor to reduce the dose of one or more of my medicines'.

**Concerns about stopping questions:**
'I have had a bad experience when stopping a medicine before'.
'I would be reluctant to stop a medicine that I had been taking for a long time'.
'If one of my medicines was stopped I would be worried about missing out on future benefits'.
'I get stressed whenever changes are made to my medicines'.
'If my doctor recommended stopping a medicine I would feel that he/she was giving up on me'.
rPATD, revised Patient Attitudes Towards Deprescribing.

---

The extensive research into the barriers and facilitators of deprescribing has shown mixed results. Older adults often hold ambivalent attitudes in that they may express a willingness to reduce their medications while perceiving all their medications as beneficial and necessary.[7 8] Clinicians can perceive their patients are reluctant to have their medications deprescribed.[9 10] A recent study from Switzerland found a quarter of patients (22/87) declined their general practitioner's (GP's) offer to deprescribe a medication in a cluster-randomised study—even with a shared decision-making intervention.[11] Similarly, a substantial proportion of participants (42%–75%) decline to participate in deprescribing intervention studies.[12–14]

To understand patients' attitudes towards deprescribing, researchers have turned towards self-reported assessments such as the Patient Perceptions of Deprescribing survey[15] and the revised Patients' Attitudes Towards Deprescribing (rPATD) questionnaire.[16] The rPATD has high uptake in the deprescribing literature with the global question most frequently used: 'If my doctor said it was possible, I would be willing to stop one or more of my medicines'. A systematic review of this questionnaire (and related versions) found inconsistency in whether there was statistical significance (and direction of the association) between characteristics and agreement with deprescribing.[17] However, mostly cross-sectional surveys were included, and few studies have used the rPATD in longitudinal research or investigating medication-related outcomes such as appropriateness or implementation of deprescribing.

It remains to be seen if patients' willingness to have medications deprescribed is associated with the implementation of actual deprescribing decisions and real changes in medication-related outcomes over time.

To address this gap in the deprescribing literature, we aimed to investigate the association between older adults' agreement with deprescribing and (1) actual change in their medications at 1 year follow-up, (2) change in the

---

**Table 1  Assessment of medication-related outcomes**

|  | Outcome | Description | Measurement |
|---|---|---|---|
| Aim 1 | Number of long-term medications | Integer number of medications prescribed for ≥90 days, based on prescribing information from electronic health records. | Change in the number of long-term medications (≥90 days, 'as needed' medications were excluded) between baseline and the 12 month follow-up. |
| Aim 2 | Medication appropriateness | The Medication Appropriateness Index (MAI)[26] is an implicit tool for assessing the appropriateness of medication prescribing. The 10-item version of the MAI was used as one of the coprimary outcomes of the OPTICA trial,[20] however, the cost effectiveness item was excluded for feasibility reasons. Using data on medications, diagnoses and laboratory values the blinded assessors rated the nine remaining criteria of the MAI for each medication prescribed for ≥90 days using a three-point scale ranging from A=appropriate, B=marginally appropriate, to C=inappropriate. | Change in the average medication appropriateness between baseline and the 12 month follow-up. We first calculated the average MAI for the baseline and the 12 month follow-up by dividing the total MAI score of the respective timepoint by the number of long-term medications at this timepoint. Then, we calculated the change in the average MAI between baseline and the 12 month follow-up. |
| Aim 3 | Implementation of prescribing recommendations to stop medications | Recommendation implemented yes vs no, as reported by GPs. | Binary variable describing whether any deprescribing recommendation to stop a medication generated by the electronic decision support system tested in the OPTICA trial had been implemented or not at the patient level. Only data from the OPTICA intervention group was used for this aim. |

MAI, Medication Appropriateness Index; OPTICA, Optimising PharmacoTherapy In the Multimorbid Elderly in Primary CAre.

**Table 2** Baseline characteristics of study participants by willingness to deprescribe (n=298)

| | All patients in the sample (n=298)* | Patients with lower than median willingness to deprescribe (n=74)† | Patients with equal or higher than median willingness to deprescribe (n=224)† |
|---|---|---|---|
| Age (in years) | 78 (74–83) | 79 (74–83) | 78 (74–83) |
| Female | 133 (45) | 39 (53) | 94 (42) |
| Patient education | | | |
| Mandatory schooling or less | 113 (38) | 25 (34) | 86 (39) |
| Diploma at secondary school level | 139 (47) | 33 (45) | 106 (47) |
| Higher education diploma | 45 (15) | 16 (22) | 29 (13) |
| Number of chronic conditions | 7 (5–10) | 7 (5–9) | 7 (5–11) |
| Living situation | | | |
| In apartment/house without any external help | 227 (76) | 62 (84) | 165 (74) |
| In apartment/house with some external help | 61 (20) | 9 (12) | 52 (23) |
| In a nursing home | 10 (3) | 3 (4) | 7 (3) |
| Patient is unable to leave the house (as compared with not housebound) | 7 (2) | 2 (3) | 5 (2) |
| Equal or higher than median satisfaction with current medication use (as compared with lower than median medication willingness to deprescribe) | 215 (72) | 59 (80) | 156 (70) |
| Number of GP consultations during the 6 month follow-up period prior to the enrolment into the study trial | 8 (5–14) | 9 (6–13) | 8 (5–15) |
| Average Medication Appropriateness Index at baseline | 1.7 (0.2–5) | 1.8 (0.2–6) | 1.7 (0.2–4.7) |
| Number of long-term medications | 8 (5–11) | 8 (5–10) | 8 (5–11) |

For continuous variables, the median and the IQR are presented. For categorical variables, frequencies and percentages are presented. Missingness: gender and age had 0% missing values. Patient education, living situation, housebound yes/no, patient satisfaction with medications and the number of chronic medications had less than 3% missing information. The number of chronic conditions and the average Medication Appropriateness Index at baseline had less than 7% missing.

*Among the 298 patients, 146 patients were then randomised to the control group and 152 patients to the intervention group. Among patients with lower than median willingness to deprescribe, 36 were randomised to the control group and 38 to the intervention group. Among patients with equal or higher than median willingness to deprescribe, 110 were assigned to the control group and 114 were randomised to the intervention group.

†Patients' willingness to have medications deprescribed was measured using the rPATD global question 'If my doctor said it was possible, I would be willing to stop one or more of my regular medicines'. The median willingness to have medications deprescribed corresponded to 'strongly agree' with the rPATD global question.

GP, general practitioner; rPATD, revised Patient Attitudes Towards Deprescribing.

appropriateness of medications at 1 year follow-up and (3) actual implementation of prescribing recommendations generated by an electronic decision support system tested in a clinical trial (OPTICA) to stop medications.

## METHODS
### Overview of the OPTICA trial
The methods and results of the 'Optimising Pharmaco-Therapy In the multimorbid elderly in primary Care' (OPTICA) trial have been reported elsewhere.[18–20] In brief, 323 patients from 43 GP practices were recruited into this cluster randomised clinical trial between January 2019 and February 2020. The 12 month follow-up ended in February 2021. 21 GPs with 160 patients were assigned to the intervention group and 22 GPs with 163 patients to the control group. Eligible patients were 65 years or older, had ≥3 chronic conditions and were taking ≥5

medications regularly. Hypothetical agreement with deprescribing was assessed at baseline. While GPs in the control group continued to provide usual care to their patients including a discussion of patients' medications in accordance with their usual practice, GPs in the intervention group performed a structured medication review centred around an electronic clinical decision support system called the 'Systematic Tool to Reduce Inappropriate Prescribing'-Assistant (STRIP-Assistant). This tool is based on the STOPP/START criteria and generated prescribing recommendations to stop, start or adapt the dosage and flagged interactions.[21–23] The OPTICA trial had a pragmatic design with data collected from participants' electronic health records (eg, medications and diagnoses) and from participants or their legal representatives over the phone (eg, quality of life, living situation, etc). The two primary outcomes of the trial were

**Table 3** Patients' attitudes towards having medications deprescribed at baseline* (n=298)

| rPATD global question: 'If my doctor said it was possible, I would be willing to stop one or more of my regular medicines' frequency (per cent) | | | | |
|---|---|---|---|---|
| Strongly agree | Agree | Unsure | Disagree | Strongly disagree |
| 224 (75) | 38 (13) | 9 (3) | 14 (5) | 13 (4) |
| Alternative measurements of patients' willingness to have medications deprescribed based on the rPATD | | | | |
| Concerns about stopping score | Mean (SD) | Median (IQR) | | |
| | 1.8 (0.8) | 1.6 (1–2.4) | | |
| 'I would like to try stopping one of my medicines to see how I feel without it' | | | | |
| Strongly agree | Agree | Unsure | Disagree | Strongly disagree |
| 120 (40) | 65 (22) | 19 (6) | 59 (20) | 35 (12) |
| 'I would like my doctor to reduce the dose of one or more of my medicines' | | | | |
| Strongly agree | Agree | Unsure | Disagree | Strongly disagree |
| 153 (51) | 64 (22) | 24 (8) | 29 (10) | 28 (9) |

Missingness: There was 0% missingness in rPATD questions and the concerns of stopping score.
*As measured by the 'revised Patients' Attitudes Towards Deprescribing (rPATD) questionnaire. Source: Reeve et al[16] 2016.
rPATD, revised Patient Attitudes Towards Deprescribing.

the improvement in the Medication Appropriateness Index (MAI) and the Assessment of Underutilisation (AOU) at 12 months.[24–26] Secondary outcomes included the number of medications, number of falls and fractures and quality of life. In the intention-to-treat analysis at 12 months, there were no group differences in the improvement of medication appropriateness (OR=1.05; 95% CI=0.59 to 1.87) nor the number of prescribing omissions (OR=0.90; 95% CI=0.41 to 1.96). The per-protocol analysis showed no statistically significant group difference either and there were no group differences in the secondary outcomes. In 59% of participants, at least one prescribing recommendation to stop or start a medication was implemented. It is of note that not all prescribing recommendations generated by STRIPA were accepted by GPs and discussed with patients. The OPTICA trial was approved by the Cantonal Ethics Committee of the Canton of Bern (*BASEC-ID: 2018–00914*). All participants or their legal representatives provided written informed consent.

### Study design and sample definition
This is a longitudinal, post-hoc sub-study of data collected during the OPTICA trial. Data from the trial baseline, the 6 month follow-up and the 12 month follow-up were used for the present analyses. This manuscript adheres to the Strengthening the Reporting of Observational Studies in Epidemiology (STROBE) reporting guideline

for observational studies.[27] All 323 participants of the OPTICA trial were older adults (≥65 years of age), with multimorbidity (≥3 chronic conditions) and polypharmacy (≥5 medications). We limited the present analyses to the participants for whom the patient version of the 'revised Patient Attitudes Towards Deprescribing' (rPATD) was used (n=298) (as compared to the caregiver version, which had been used for patients included by proxy consent of their legal representative).[16]

### Assessment of patients' agreement with deprescribing
Patients' attitudes towards hypothetical deprescribing were measured using the rPATD at baseline. The rPATD contains 22 questions with 'Strongly disagree'[1] and 'Strongly agree'[5] as the scale anchors.[16] For the main analyses, we used the global question from the rPATD 'If my doctor said it was possible, I would be willing to stop one or more of my regular medicines' as the independent variable, which measures patients' level of agreement with accepting deprescribing if it were proposed by a medical doctor. In addition, we used two questions from the rPATD 'appropriateness' factor ('I would like to try stopping one of my medicines to see how I feel without it' and 'I would like my doctor to reduce the dose of one or more of my medicines'), which aim to measure patients' agreement to try to stop or reduce medicines (box 1). Furthermore, we used the rPATD 'concerns about stopping' factor score (ranging from 1 to 5) calculated based on five rPATD questions as independent variables. Several questions from the rPATD were used as independent variables given the ceiling effect of the global rPATD willingness to deprescribe question and the fact that there is more variation in the responses to the other two rPATD questions and the concerns about stopping score.

### Assessment of medication-related outcomes over time
Medication-related outcomes over time were assessed using data collected at baseline and throughout the follow-up period of the OPTICA trial. Details on how the three medication-related outcomes were assessed—change in the number of medications, medication appropriateness and the implementation of prescribing recommendations—can be found in table 1.

### Covariates
The following variables were used to adjust the analyses: gender, age, educational status, number of chronic conditions, living situation, capable of leaving the house (yes/no), patients' satisfaction with medications and number of GP visits in the 6 months prior to the study enrolment. The included variables were based on the literature of the factors associated with number of medications/polypharmacy and the factors associated with potentially inappropriate medication use/medication appropriateness considering the data available from the OPTICA trial.[28–34]

### Statistical analysis
First, we described the demographics and main clinical characteristics of the study participants. Second, we

**Table 4** Multivariate associations between the change in the number of medications throughout the 12 month follow-up period and patients' attitudes towards deprescribing (n=253)

| Name of the variable | Coefficient | P value | 95% CI |
|---|---|---|---|
| rPATD global question: 'If my doctor said it was possible, I would be willing to stop one or more of my regular medicines' (reference: strongly agree) | | | |
| Agree | −0.96 | 0.169 | −2.33 to 0.41 |
| Unsure | 0.61 | 0.963 | −2.52 to 2.64 |
| Disagree | 0.58 | 0.598 | −1.56 to 2.71 |
| Strongly disagree | 0.26 | 0.806 | −1.81 to 2.33 |
| Alternative measurements of patients' willingness to have medications deprescribed based on the rPATD | | | |
| Concerns about stopping score (per 1-unit increase)* | | | |
| | 0.65 | **0.026*** | 0.08 to 1.22 |
| 'I would like to try stopping one of my medicines to see how I feel without it' (reference: strongly agree) | | | |
| Agree | −0.12 | 0.830 | −1.33 to 1.07 |
| Unsure | 0.62 | 0.509 | −1.24 to 2.51 |
| Disagree | 0.47 | 0.448 | −0.74 to 1.68 |
| Strongly disagree | −0.21 | 0.774 | −1.68 to 1.25 |
| 'I would like my doctor to reduce the dose of one or more of my medicines' (reference: strongly agree) | | | |
| Agree | 1.13 | 0.070 | −0.09 to 2.36 |
| Unsure | −0.97 | 0.251 | −2.64 to 0.69 |
| Disagree | 0.79 | 0.306 | −0.72 to 2.31 |
| Strongly disagree | 0.71 | 0.359 | −0.81 to 2.24 |

Multilevel linear regression models adjusted for patient age, education status, gender, number of chronic conditions, living situation, whether the patient is housebound or not, patient satisfaction with medications, the number of GP consultations in the 6 months prior to the study inclusion and the group allocation during the trial. Missingness: the change in the number of chronic medications over the 12 month follow-up period had 8% missing values. / *<0.05
*As calculated based on Reeve et al[16] 2016.
GP, general practitioner; rPATD, revised Patient Attitudes Towards Deprescribing.

descriptively analysed three questions from the rPATD and the concerns about stopping score to describe patients' attitudes towards at baseline. Third, we performed a set of multilevel regression analyses. For aims 1 and 2, we used multilevel linear regression models to investigate the association between patients' agreement with deprescribing and the outcomes. In subgroup analyses, we restricted the analyses to the OPTICA intervention group. For aim 3, we used a multilevel logistic regression model to investigate the association between patients' agreement with deprescribing and the binary outcome variable. For aim 3, we used data from the OPTICA intervention group only. All analyses were adjusted for the clustering effect at the GP level and the measurable covariates listed in the section above plus the group allocation during the trial (except for the analyses for aim 3, which were based on data from the intervention group only). Analyses were limited to the observed data, and we did not use any multiple imputation methods. All analyses were performed with STATA V.15.1 (StataCorp, College Station, TX, USA). A p-value<0.05 was considered significant.

### Patient and public involvement

No patients were involved in this sub-study of the OPTICA trial.

### RESULTS

### Baseline characteristics

Table 2 describes the baseline characteristics of study participants. Out of the 298 participants for whom information on their attitudes towards deprescribing was assessed at baseline (92% of all trial participants), 45% were women and the median age was 78 years; 75% (224/298) of the participants had equal or higher than median agreement with deprescribing as measured by the global rPATD question.

### Proxy measures for patients' attitudes towards deprescribing

Table 3 shows the descriptive results of the different measures used to assess patients' willingness to have medications deprescribed. More than 85% of participants strongly agreed or agreed with the rPATD global question and only 9% of participants disagreed with this statement, whereas there was slightly more variation in responses to the other two rPATD questions. Approximately 60% of participants reported that they would like to try stopping one of their medications to see how they would feel without it, whereas 32% disagreed or strongly disagreed with this statement.

Table 5    Multivariate associations between the change in the medication appropriateness* throughout the 12 month follow-up period and patients' attitudes towards deprescribing † (n=242)

| Name of the variable | Coefficient | P value | 95% CI |
|---|---|---|---|
| rPATD global question: 'If my doctor said it was possible, I would be willing to stop one or more of my regular medicines' (reference: strongly agree) | | | |
| Agree | 0.35 | 0.426 | −0.51 to 1.21 |
| Unsure | 0.92 | 0.293 | −0.79 to 2.63 |
| Disagree | −1.01 | 0.145 | −2.36 to 0.35 |
| Strongly disagree | −0.80 | 0.221 | −2.08 to 0.48 |
| Alternative measurements of patients' willingness to have medications deprescribed based on the rPATD | | | |
| Concerns about stopping score (per 1-unit increase)* | | | |
| | −0.29 | 0.120 | −0.65 to 0.08 |
| 'I would like to try stopping one of my medicines to see how I feel without it' (reference: strongly agree) | | | |
| Agree | −0.45 | 0.253 | −1.21 to 0.32 |
| Unsure | −0.66 | 0.281 | −1.87 to 0.54 |
| Disagree | −0.45 | 0.246 | −1.22 to 0.31 |
| Strongly disagree | −0.57 | 0.233 | −1.51 to 0.37 |
| 'I would like my doctor to reduce the dose of one or more of my medicines' (reference: strongly agree) | | | |
| Agree | −0.44 | 0.253 | −1.20 to 0.32 |
| Unsure | −0.59 | 0.282 | −1.67 to 0.49 |
| Disagree | −0.02 | 0.968 | −0.95 to 0.99 |
| Strongly disagree | 0.13 | 0.795 | −0.85 to 1.11 |

Multilevel linear regression models adjusted for patient age, education status, gender, number of chronic conditions, living situation, whether the patient is housebound or not, patient satisfaction with medications, the number of GP consultations in the 6 months prior to the study inclusion, and the group allocation during the trial. Missingness: The change in the Medication Appropriateness Index over the 12 month follow-up period had 13% missing values.
*As assessed using the Medication Appropriateness Index: Samsa et al[26] 1994.
†As calculated based on Reeve et al[16] 2016.
rPATD, revised Patient Attitudes Towards Deprescribing.

## Number of medications over time

Table 4 shows the associations between the different measures assessing patients' attitudes towards deprescribing and the change in number of medications throughout the 12 month follow-up period. At the 12 month follow-up, the mean change in the number of medications was −0.2 (SD=4.2). The only statistically significant association was between the concerns about stopping score and the change in the number of medications (coefficient: 0.65, 95% CI: 0.08 to 1.22). A higher score indicates being more concerned about stopping. Meaning, per 1-unit increase in the concerns about stopping score the change in the number of medications between baseline and the 12 month follow-up increased by 0.65. In the subgroup analyses restricted to the OPTICA intervention group, we found evidence for a statistically significant association between patients' concerns about stopping score and an increase in the number of medications between baseline and the 12 month follow-up period (coefficient: 0.78, 95% CI: 0.04 to 1.52, p-value: 0.04 - other data not presented).

## Medication appropriateness over time

The associations between patients' willingness to have medications deprescribed and the change in medication appropriateness throughout the 12 month follow-up period are shown in table 5. At the 12 month follow-up, the mean change in the average Medication Appropriateness Index was −0.75 (SD=2.5). We did not find evidence for any statistically significant association. In the subgroup analyses restricted to the OPTICA intervention group, we found evidence for a statistically significant association between patients' being undecided or (strongly) agreeing with the statement 'I would like my doctor to reduce the dose of one or more of my medicines' and an improvement in medication appropriateness between baseline and the 12 month follow-up period (undecided: coefficient: -2.02, 95% CI: -3.87 to -0.18, p-value: 0.03 / agree: -1.99, 95% CI: -3.41 to -0.58, p-value: 0.006 / strongly agree: coefficient: -1.60, 95% CI: -2.89 to -0.31, p-value: 0.015 - other data not presented).

## Implementation of prescribing recommendations

Table 6 shows the association between patients' attitudes towards deprescribing and the implementation of prescribing recommendations that were generated as part of the OPTICA medication review intervention (n=31). On average, one prescribing recommendation to stop or start a medication were reported to be implemented per

**Table 6** Multivariate associations between the implementation of recommendations to stop medications and patients' attitudes towards deprescribing * (n=31)

| Name of the variable | OR | P value | 95% CI |
|---|---|---|---|
| rPATD global question: 'If my doctor said it was possible, I would be willing to stop one or more of my regular medicines' (reference: equal or higher than median agreement)† | | | |
| Below median agreement | 4.90 | 0.244 | 0.34 to 71.3 |
| Alternative measurements of patients' willingness to have medications deprescribed based on the rPATD | | | |
| Concerns about stopping score (per 1-unit increase) | | | |
| | 1.13 | 0.812 | 0.41 to 3.13 |
| 'I would like to try stopping one of my medicines to see how I feel without it' (reference: equal or higher than median agreement)† | | | |
| Below median agreement | 2.53 | 0.305 | 0.43 to 14.89 |
| 'I would like my doctor to reduce the dose of one or more of my medicines' (reference: equal or higher than median agreement)† | | | |
| Below median agreement | 7.82 | 0.086 | 0.75 to 82.2 |

The analyses presented in this table used data from the OPTICA intervention group only. Despite several reminders, only a couple of GPs from the OPTICA intervention group reported this information.
*Multilevel logistic regression models adjusted for patient age and gender.
†Due to the low number of observations for which the implementation of recommendations was reported, the rPATD question was dichotomised.
GP, general practitioner; OPTICA, Optimising PharmacoTherapy In the Multimorbid Elderly in Primary CAre.

patient (reported elsewhere[35]) and 59% of patients in the intervention group had one or more prescribing recommendation implemented. We did not find any evidence for a statistically significant association between patients' attitudes towards deprescribing and the implementation of deprescribing recommendations.

## DISCUSSION

In this sub-study of a cluster randomised clinical trial, we examined the association between older adults' attitudes towards deprescribing and change in a participant's medications, appropriateness of their medications and actual implementation of prescribing recommendations. Overall, we found that these medication-related outcomes measured over 1 year did not seem to be associated with the rPATD deprescribing questions measured in this study. To consider reasons why no association was found, first we discuss the rPATD questions in more detail and their ability to measure self-reported attitudes towards deprescribing. Second, consideration is given to our medication optimization intervention and how medication-related outcomes were measured in this study.

In our study, 88% of participants agreed or strongly agreed with the rPATD global question: 'If my doctor said it was possible, I would be willing to have one or more of my medications deprescribed'. However, this high agreement was not associated with changes in medication-related outcomes over time. Other deprescribing intervention trials using the rPATD global question[36–38] also found high agreement with hypothetical deprescribing (86%–95%) with no effect on deprescribing or medication-related outcomes. The majority of studies using the rPATD global deprescribing question report greater than 80% agreement;[8] therefore, it may be

difficult to find an association given the ceiling effect of this question.

A recent cluster randomised controlled trial conducted in Ireland with older adults taking ≥15 medications found that a higher agreement with deprescribing measured by the rPATD was not only associated with a higher rate of deprescribing but also initiating medicines.[17] The authors note that the rPATD global question may identify participants who are agreeable to any medication-changes if they are suggested by their doctor. Supporting this, there is variation between the global question and another rPATD deprescribing question which does not refer to the doctor.[8] In our study, agreement was much higher for the rPATD global deprescribing question with 88% of participants willing to deprescribe if their doctor said it was possible, however 62% wanted to try stopping one of their medications to see how they would feel without it. Other studies using the rPATD have shown substantial differences (30%–73% gap) between these questions with the global question responses always higher[8 39–43] suggesting the influence of the doctor should not be underestimated. Similarly, a content analysis including over 2500 participants from Australia, the UK and the USA found that approximately one-half of older adults who agreed with deprescribing in a hypothetical scenario felt that the doctor's recommendation was an important consideration.[44]

There is a complex interplay of factors, such as clinical decision-making and patients' attitudes, that are behind acceptance (or not) of deprescribing. It is possible that the lack of association between patients' agreement with deprescribing and medication-related changes in our study was due to the inconclusive effectiveness of the OPTICA deprescribing intervention, which is similar

to other deprescribing/medication optimization interventional studies. While it is useful to quantify attitudes towards deprescribing to get a sense of older adults' general thoughts about their medications, it may be unfair to expect self-reported attitudes to equate to actual medication changes in the absence of an effective intervention. Of note, an exploratory deprescribing controlled trial conducted in Australia measured the original PATD (10 questions) at baseline and again at follow-up.[45] Although the PATD baseline scores did not predict deprescribing outcomes, statistically significant changes were shown in three questions which signalled a shift in patients' beliefs about medicines following a deprescribing intervention.

Deprescribing in clinical practice and interventional studies may not occur for many reasons, such as if the GP chooses not to initiate it. From the main OPTICA trial, the most common reasons for not implementing prescribing recommendations were that GPs thought that patients' current medications were beneficial and that the recommended change was not suitable. The first study to focus on older adults from multiple countries who disagree with a deprescribing recommendation in a vignette-based survey (n=899)[46] found that older adults reported valuing their medications, they expressed doubts about deprescribing, and preferred to avoid change. Respondents who disagreed with the deprescribing recommendation, as opposed to those who strongly disagreed, were more interested in alternative strategies such as improved communication or a replacement medication. Further to this, respondents reported different factors for disagreeing with a deprescribing recommendation based on the medication type (lansoprazole vs simvastatin). Taken together, attitudinal measures of deprescribing may benefit from greater sensitivity to reluctance towards deprescribing, less vulnerability to the doctor's influence and capturing attitudes towards specific medications.[43 47] Ultimately, it would be useful for a tool to identify patients at different degrees of willingness to deprescribe so that deprescribing interventions can be tailored to their needs and preferences. Further exploration is needed into the link between attitudes towards medicines and actual medication changes, possibly through process evaluations of deprescribing trials.

### Strengths and limitations

The present analyses were strengthened by the longitudinal design, which allows for a clear temporal distinction between patients' attitudes towards deprescribing assessed at baseline and the medication-related outcomes over time. We would like to emphasise the following limitations of these present analyses. First, patients agreeing to participate in the OPTICA trial could have had a greater interest in their medications than those who chose not to participate and potentially a higher willingness to have one or more of their medications deprescribed. Some patients were excluded from the analyses due to missing data on their medication. Also, to determine the medication-related outcomes for aims 1 and 2, we used prescribing data from electronic health records, which does not necessarily correspond to what medications were actually used by patients. Finally, despite several reminders, only a small proportion of GPs from the intervention group reported which prescribing recommendations were implemented together with patients. This explains the smaller sample size for aim 3. Due to the small sample size used to analyse aim 3, the confidence intervals were wide and imprecise.

### CONCLUSIONS

Our findings indicate that there does not seem to be an association between most measures of patient agreement with deprescribing and medication-related outcomes over time. It is important to capture a range of participant attitudes that are both for and against deprescribing, as well as to consider the relationship between self-report surveys and actual deprescribing. The results highlight the need for further research to better understand the factors that contribute to successful deprescribing in primary care settings.

**Author affiliations**
[1]Institute of Primary Health Care BIHAM, University of Bern, Bern, Switzerland
[2]Center for Healthcare Delivery Sciences (C4HDS), Division of Pharmacoepidemiology and Pharmacoeconomics, Department of Medicine, Brigham and Women's Hospital and Harvard Medical School, Boston, MA, USA
[3]Sydney School of Public Health, Faculty of Medicine and Health, The University of Sydney, Sydney, New South Wales, Australia
[4]Community Pharmacy, Centre for Primary Care and Public Health (Unisanté), University of Lausanne, Lausanne, Switzerland
[5]School of Pharmaceutical Sciences, University of Geneva, Geneva, Switzerland

**Acknowledgements** We would like to thank the general practitioners who participated in the OPTICA trial for their great contributions without which it would have been impossible to conduct this study. We also thank the participants for consenting to participate in our study. We would like to thank Dr Wade Thompson for his guidance on the manuscript. Dr Jungo was a member of the Junior Investigator Intensive Program of the US Deprescribing Research Network, which is funded by the National Institute on Aging.

**Contributors** All authors contributed to the concept and study design. All authors contributed to the acquisition, analysis or interpretation of data. KTJ and KRW wrote the first draft of the manuscript. All other authors provided feedback and approved the final version of the manuscript. KTJ and SS provided administrative and technical support. KTJ did the statistical analyses. SS obtained funding for the work and supervised the study. KTJ and KRW had full access to all the data in the study and are the guarantors.

**Funding** The OPTICA trial was supported by the Swiss National Science Foundation, within the framework of the National Research Programme 74 'Smarter Health Care' (NRP74) under contract number 407440_167465 (to Prof SS, Prof Rodondi and Prof Schwenkglenks). Dr KTJ was funded by a Postdoc.Mobility Fellowship from the Swiss National Science Foundation (P500PM_206728). Dr KRW was funded by a Swiss Government Excellence Scholarship (2021.0684), a Swiss National Science Foundation Scientific Exchanges grant and a NHMRC Emerging Leader Research Fellowship (2017295).

**Competing interests** None declared.

**Patient and public involvement** Patients and/or the public were not involved in the design, or conduct, or reporting or dissemination plans of this research.

**Patient consent for publication** Not required.

**Ethics approval** This study involves human participants and was approved by Cantonal Ethics Committee of the Canton of Bern (BASEC-ID: 2018–00914) Participants gave informed consent to participate in the study before taking part.

**Provenance and peer review** Not commissioned; externally peer reviewed.

**Data availability statement** Data are available upon reasonable request. We will make the data for this study available to other researchers upon request after publication. The data will be made available for scientific research purposes, after the proposed analysis plan has been approved. Data and documentation will be made available through a secure file exchange platform after approval of the proposal. In addition, a data transfer agreement must be signed (which defines obligations that the data requester must adhere to with regard to privacy and data handling). Deidentified participant data limited to the data used for the proposed project will be made available, along with a data dictionary and annotated case report forms. For data access, please contact the corresponding author.

**ORCID iDs**
Katharina Tabea Jungo http://orcid.org/0000-0002-1782-1345
Kristie Rebecca Weir http://orcid.org/0000-0002-9507-5050
Damien Cateau http://orcid.org/0000-0002-6540-9160

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
