## [Reviewer comments · BMJ Open]

ARTICLE DETAILS

TITLE (PROVISIONAL)	Older adults' willingness to deprescribe and medication changes: A longitudinal sub-study of a cluster randomized controlled trial
AUTHORS	Jungo, Katharina Tabea; Weir, Kristie; Cateau, Damien; Streit, Sven

VERSION 1 – REVIEW

REVIEWER	Sion Scott University of Leicester College of Life Sciences, School of Healthcare
REVIEW RETURNED	09-Jun-2023

GENERAL COMMENTS	Thanks for inviting me to review this longitudinal sub-study of the OPTICA trial linking rPATD data with medicines outcomes. The manuscript is very well written and methods are robust. I have only a few comments to polish. Abstract - not sure why it mentions rPATD at baseline without also mentioning follow-up. I think you mean it was given alongside OPTICA baseline outcome data collection. Perhaps just rephrase so its clear there isn't a follow-up to 'follow'. Results Table 2 - it's quite hard to follow with the Likert scales presented vertically. Could you have them going across e.g. in other rAPTD papers: https://jamanetwork.com/journals/jamainternalmedicine/fullarticle/2706177 table 1 Discussion Its worth mentioning that the lack of an association could also be because our interventions are not effective. Patients can have high willingness (i.e. the rPATD could be valid) and are thus a captive audience but will still have barriers to address before saying yes that need to be addressed by clinicians and by extension interventions. The qual literature says that inherently patients do not want to take medicines they don't need, which supports the high rPATD scores. Was there anything in the OPTICA process evaluation to comment on this? Either why, what could this gap be due to? Also, what you've done here is useful and I wonder whether one of your recommendations should be that more trials do this so that we can add to the evidence regarding the link (or lack of) between willingness and outcomes, and possibly types of interventions and whether this matters.
--

REVIEWER	Mary McPherson University of Maryland Baltimore, PPS
REVIEW RETURNED	23-Jun-2023

GENERAL COMMENTS	Thanks for the opportunity to review your manuscript! Just a few comments: Page 6 - inclusion criteria included taking 5 or more meds regularly - I assuming you did NOT include "as needed" medications Page 6 - the STRIP-Assistant tool - was this developed by these authors? Was it validated? Page 6 - might be useful to reader to understand a bit better the "Medication Appropriateness Index" and "Assessment of Underutilization" tools (without having to pull the references) Page 7 - re: the OPTICA trial - I assume this was published - is this ref 20? Should you include that ref here? Page 8 - you may want to define what 1 = and 5 = in the 1-5 Likert response scale Page 9 - it might be nice to include justification for all your covariates. Is there literature support stating each variable can influence outcomes? Page 11 - Just making sure I understand this correctly - per 1-unit increase in the concerns about stopping score (and the higher the score, the more reluctant to change) is correlated with the change in number of meds from baseline to 12 mo follow up (meaning every unit higher [less excited about med changes] correlated with one additional medication prescribed at 12 months. Right? Last, just confirming there was no attempt to query the PRESCRIBER's interest in deprescribing, correct? The patient can say yay or nay all day long, but the prescriber is the one who pulls the trigger! Thanks.
---

VERSION 1 – AUTHOR RESPONSE

Reviewer: 1

Dr. Sion Scott, University of Leicester College of Life Sciences

Thanks for inviting me to review this longitudinal sub-study of the OPTICA trial linking rPATD data with medicines outcomes. The manuscript is very well written and methods are robust. I have only a few comments to polish.

RESPONSE: We would like to thank the reviewer for this positive feedback.

1. Abstract - not sure why it mentions rPATD at baseline without also mentioning follow-up. I think you mean it was given alongside OPTICA baseline outcome data collection. Perhaps just rephrase so its clear there isn't a follow-up to 'follow'.

RESPONSE: Yes, that is what we mean. Reference to baseline has now been deleted from the Exposures and Results sections of the Abstract.

2. Results - Table 2 - it's quite hard to follow with the Likert scales presented vertically. Could you have them going across e.g. in other rAPTD papers:

https://jamanetwork.com/journals/jamainternalmedicine/fullarticle/2706177_table_1

RESPONSE: Thank you for this suggestion, we have updated the table accordingly.

3. Discussion - Its worth mentioning that the lack of an association could also be because our interventions are not effective. Patients can have high willingness (i.e. the rPATD could be valid) and are thus a captive audience but will still have barriers to address before saying yes that need to be addressed by clinicians and by extension interventions. The qual literature says that inherently patients do not want to take medicines they don't need, which supports the high rPATD scores. Was there anything in the OPTICA process evaluation to comment on this? Either why, what could this gap be due to?

RESPONSE: We agree the lack of association could also be due to the effectiveness of deprescribing interventions. Qualitative literature does support the notion that many older adults would prefer to take fewer medications, however, older adults also often believe their medicines are important and necessary with coexisting, contradictory beliefs about medicines. We looked into the process evaluation of the OPTICA trial, but unfortunately, we do not have any evidence explaining this gap.

We have added to the Discussion: "There is a complex interplay of factors, such as clinical decision-making and patients' attitudes, that are behind acceptance (or not) of deprescribing. It is possible that the lack of association between the rPATD and medication-related changes in our study was due to the inconclusive effectiveness of the OPTICA deprescribing intervention, which is similar to other deprescribing interventional studies. While it is useful to quantify attitudes towards deprescribing to get a sense of older adults' general thoughts about their medications, it may be unfair to expect self-reported attitudes to equate to actual medication changes."

Also added to the Discussion: "Further exploration is needed into the link between attitudes towards medicines and actual medication changes, possibly through process evaluations of deprescribing trials."

4. Also, what you've done here is useful and I wonder whether one of your recommendations should be that more trials do this so that we can add to the evidence regarding the link (or lack of) between willingness and outcomes, and possibly types of interventions and whether this matters.

RESPONSE: We fully agree with the statement that currently little is known about the association between willingness to have medications deprescribed and outcomes. If deprescribing trials consistently measured willingness to deprescribe at baseline, this would shed more light on the relationship between willingness to deprescribing and outcomes.

Added to the Discussion: "Further exploration is needed into the link between attitudes towards medicines and actual medication changes, possibly through process evaluations of deprescribing trials."

Reviewer: 2

Prof. Mary McPherson, University of Maryland Baltimore

Thanks for the opportunity to review your manuscript! Just a few comments:

1. Page 6 - inclusion criteria included taking 5 or more meds regularly - I assuming you did NOT include "as needed" medications

RESPONSE: That is correct, we did not include "as needed" medications. We have added this information to Box 2.

2. Page 6 - the STRIP-Assistant tool - was this developed by these authors? Was it validated?

RESPONSE: The Systematic Tool to Reduce Inappropriate Prescribing Assistant tool was developed by researchers at the University of Utrecht in the Netherlands. The tool has been used in different previous studies:

- Drenth-van Maanen AC, Leendertse AJ, Jansen PAF, Knol W, Keijsers CJPW, Meulendijk MC, van Marum RJ. The Systematic Tool to Reduce Inappropriate Prescribing (STRIP): Combining implicit and explicit prescribing tools to improve appropriate prescribing. *J Eval Clin Pract.* 2018 Apr;24(2):317-322. doi: 10.1111/jep.12787. Epub 2017 Aug 4. PMID: 28776873.
- Keijsers CJ, van Doorn AB, van Kalles A, de Wildt DJ, Brouwers JR, van de Kamp HJ, Jansen PA. Structured pharmaceutical analysis of the Systematic Tool to Reduce Inappropriate Prescribing is an effective method for final-year medical students to improve polypharmacy skills: a randomized controlled trial. *J Am Geriatr Soc.* 2014 Jul;62(7):1353-9. doi: 10.1111/jgs.12884. Epub 2014 Jun 10. PMID: 24916615.
- Blum MR, Sallevelt BTGM, Spinewine A, O'Mahony D, Moutzouri E, Feller M, Baumgartner C, Roumet M, Jungo KT, Schwab N, Bretagne L, Beglinger S, Aubert CE, Wilting I, Thevelin S, Murphy K, Huibers CJA, Drenth-van Maanen AC, Boland B, Crowley E, Eichenberger A, Meulendijk M, Jennings E, Adam L, Roos MJ, Gleeson L, Shen Z, Marien S, Meinders AJ, Baretella O, Netzer S, de Montmollin M, Fournier A, Mouzon A, O'Mahony C, Aujesky D, Mavridis D, Byrne S, Jansen PAF, Schwenkglens M, Spruit M, Dalleur O, Knol W, Trelle S, Rodondi N. Optimizing Therapy to Prevent Avoidable Hospital Admissions in Multimorbid Older Adults (OPERAM): cluster randomised controlled trial. *BMJ.* 2021 Jul 13;374:n1585. doi: 10.1136/bmj.n1585. Erratum in: *BMJ.* 2022 Dec 1;379:o2859. PMID: 34257088; PMCID: PMC8276068.

3. Page 6 - might be useful to reader to understand a bit better the "Medication Appropriateness Index" and "Assessment of Underutilization" tools (without having to pull the references)

RESPONSE: We have added further detail about the Medication Appropriate Index and other medication-related outcomes to Box 2. We did not include the Assessment of Underutilization tool in this sub-study so have not added detail about this outcome.

4. Page 7 - re: the OPTICA trial - I assume this was published - is this ref 20? Should you include that ref here?

RESPONSE: Thank you, this reference has now been added.

5. Page 8 - you may want to define what 1 = and 5 = in the 1-5 Likert response scale

RESPONSE: We have these numbers in brackets: "The rPATD contains 22 questions with "Strongly disagree (1)" and "Strongly agree (5)" as the scale anchors."

6. Page 9 - it might be nice to include justification for all your covariates. Is there literature support stating each variable can influence outcomes?

RESPONSE: We selected the covariates included in the analyses based on the literature of the factors associated with number of medications/polypharmacy and the factors associated with potentially inappropriate medication use/medication appropriateness and the variables that were available in the data collected during the OPTICA trial. We would like to emphasize though that the literature shows mixed results regarding some of these variables with the direction of associations varying across studies.

We have added to the manuscript: “The included variables were based on the literature of the factors associated with number of medications/polypharmacy and the factors associated with potentially inappropriate medication use/medication appropriateness considering the data available from the OPTICA trial.”

7. Page 11 - Just making sure I understand this correctly - per 1-unit increase in the concerns about stopping score (and the higher the score, the more reluctant to change) is correlated with the change in number of meds from baseline to 12 mo follow up (meaning every unit higher [less excited about med changes] correlated with one additional medication prescribed at 12 months. Right?

RESPONSE: Indeed, the coefficient can be interpreted as the expected change in the outcome (change in the number of medications) per unit change in the exposure variable (concerns about stopping score). The higher the score, the higher the number of medications at baseline.

8. Last, just confirming there was no attempt to query the PRESCRIBER's interest in deprescribing, correct? The patient can say yay or nay all day long, but the prescriber is the one who pulls the trigger! Thanks.

RESPONSE: Some data from the main OPTICA trial was collected regarding general practitioners' reasons for not implementing the prescribing implementations. We have added this point to the discussion:

“Deprescribing in clinical practice and interventional studies may not occur for many reasons, such as if the general practitioner chooses not to initiate it. From the main OPTICA trial, the most common reasons for not implementing prescribing recommendations were that general practitioners thought that patients' current medications were beneficial and that the recommended change was not suitable.”

VERSION 2 – REVIEW

REVIEWER	Mary McPherson University of Maryland Baltimore, PPS
REVIEW RETURNED	23-Sep-2023

GENERAL COMMENTS	This review was conducted in tandem with Derek Edwards, PharmD, MS, PGY-2 Pain Management and Palliative Care Pharmacy Resident at the University of Maryland School of Pharmacy. Derek has no relevant competing interests that may impact this review. Please see the attached document for the full review comments. The reviewer provided a marked copy with additional comments. Please contact the publisher for full details.
---

VERSION 2 – AUTHOR RESPONSE

We would like to thank Mr Edwards for the comprehensive and in-depth review of our manuscript. Although minor revisions, our manuscript has improved once again by the review process. We look forward to receiving the editor's decision on our manuscript. Please note there were no comments from the Editor or formatting amendments.

Reviewer: 3

Derek Edwards, School of Pharmacy, University of Maryland, Baltimore

Duplicate keywords? Add deprescribing vs. geriatrics instead? Keywords on page 2 different/more appropriate

RESPONSE: The keywords on page 2 are the ones that we selected.

Add, "years" following 78

RESPONSE: Added.

Add colon after "limitations" for homogeneity

RESPONSE: Added.

Points 1-3 seem to be more so objective statements rather than clear strengths/limitations of investigation

RESPONSE: We have adapted two of the points, so they are now clear strengths.

remove, "they" from sentence for readability

RESPONSE: Deleted.

Feeling conflicted on including results from OPTICA trial in introduction for readability sake. For further discussion (maybe remove precise OR, CI...etc. and summarize results only?)

RESPONSE: Given that we only briefly mention the results from the main study, and it is common practice to provide odds ratios and confidence intervals, we believe that our paper is overall still readable.

Switch "Sample Definition" to all lower case for homogeneity

RESPONSE: Done.

Would like additional details on how this was assessed (who assessed, was there blinding, were scores calculated by one assessor or multiple and compared?)

RESPONSE: As reported in the OPTICA trial publication [1], blinded study team members rated the 9 MAI criteria for each of the chronic medications study participants were using based on standard operating procedures developed for the assessment. The inter-rater reliability assessments showed moderate agreement regarding the MAI assessments.

[1]Jungo, KT, Ansorg A, Floriani C, Rozsnyai Z, Schwab N, Meier R et al. Optimising prescribing in older adults with multimorbidity and polypharmacy in primary care (OPTICA): cluster randomised clinical trial BMJ 2023; 381 :e074054 doi:10.1136/bmj-2022-074054

Additional details needed on p-value for significance?

RESPONSE: We added the following sentence "A p-value < 0.05 was considered significant" to the manuscript.

Remove grey line below this row (Table 3 and 4)

RESPONSE: In our version, there is no line here. If it remains, this will be fixed at the copy-editing stage.

Too many tables/boxes/figures included in article? Per:

https://bmjopen.bmj.com/pages/authors#submission_guidelines, recommend 5 figures or less (flexible).

RESPONSE: Thank you for checking the guidance, they state: "Word count, we recommend... up to five figures and tables. This is flexible, but exceeding this will impact upon the paper's 'readability'." We have 5 tables although they are short and not over many lines or pages. We do not think that our number of tables or boxes impacts the readability of the manuscript.

Update formatting to, "rPATD" in line with rest of paper

RESPONSE: Updated.

Sentence is a bit long and difficult to digest

RESPONSE: We have split this sentence to improve clarity. "The present analyses were strengthened by the longitudinal design, which allows for a clear temporal distinction between patients' willingness to have medications deprescribed assessed at baseline and the medication-

related outcomes over time. Additionally, the intervention to optimize medication was offered randomly.”

This sentence is a bit vague, could benefit from additional specification (or re-wording)

RESPONSE: Original sentence: “Due to challenges with how data from the electronic health records of participating patients were recorded, there was some missing data on medication, which is why some participants were excluded from the analyses.”

This was a bit vague, we have adjusted the sentence: “Some patients were excluded from the analyses due to missing data on their medication.”

Formatting off (repeated words in citation?)

RESPONSE: Thank you for spotting this, we have fixed it.